# Enhanced Corrosion Resistance and Local Therapy from Nano-Engineered Titanium Dental Implants

**DOI:** 10.3390/pharmaceutics15020315

**Published:** 2023-01-17

**Authors:** Tianqi Guo, Jean-Claude Scimeca, Sašo Ivanovski, Elise Verron, Karan Gulati

**Affiliations:** 1School of Dentistry, The University of Queensland, Herston, QLD 4006, Australia; 2CNRS, Inserm, iBV, Université Côte d’Azur, 06108 Nice, France; 3CNRS, CEISAM, UMR 6230, Nantes Université, 44000 Nantes, France

**Keywords:** nano-engineering, dental implants, titanium, drug delivery, corrosion, corrosion resistance

## Abstract

Titanium is the ideal material for fabricating dental implants with favorable biocompatibility and biomechanics. However, the chemical corrosions arising from interaction with the surrounding tissues and fluids in oral cavity can challenge the integrity of Ti implants and leach Ti ions/nanoparticles, thereby causing cytotoxicity. Various nanoscale surface modifications have been performed to augment the chemical and electrochemical stability of Ti-based dental implants, and this review discusses and details these advances. For instance, depositing nanowires/nanoparticles via alkali-heat treatment and plasma spraying results in the fabrication of a nanostructured layer to reduce chemical corrosion. Further, refining the grain size to nanoscale could enhance Ti implants’ mechanical and chemical stability by alleviating the internal strain and establishing a uniform TiO_2_ layer. More recently, electrochemical anodization (EA) has emerged as a promising method to fabricate controlled TiO_2_ nanostructures on Ti dental implants. These anodized implants enhance Ti implants’ corrosion resistance and bioactivity. A particular focus of this review is to highlight critical advances in anodized Ti implants with nanotubes/nanopores for local drug delivery of potent therapeutics to augment osseo- and soft-tissue integration. This review aims to improve the understanding of novel nano-engineered Ti dental implant modifications, focusing on anodized nanostructures to fabricate the next generation of therapeutic and corrosion-resistant dental implants. The review explores the latest developments, clinical translation challenges, and future directions to assist in developing the next generation of dental implants that will survive long-term in the complex corrosive oral microenvironment.

## 1. Introduction

Based on the concept that the living bone can directly form biological interconnection with a Ti surface at the microscopic level (osseointegration), Ti has been regarded as the material of choice for fabricating endosseous implants [1,2]. Ti has been widely utilized to fabricate dental implants and orthopedic fixation plates/screws to replace missing teeth and stabilize the bone fracture, respectively, attributed to its favorable mechanical strength and biocompatibility [3]. Furthermore, it has been reported that high survival rates (96~98% over 10 years) of Ti-based dental implants could be achieved when performed by adequately trained physicians [4]. Despite their high survival rates, Ti-based dental implants are constantly challenged by mechanical stimulation and chemical corrosion within the corrosive oral cavity environment [5,6]. A protective oxide layer (TiO_2_) readily forms on Ti upon oxygen exposure, shielding the underlying implant against mechanical, chemical, and electrochemical challenges [1]. However, such native oxide layers are thin (2~5 nm thick), amorphous, and have many defects that can expose the underlying Ti to a corrosive environment [1].

In response to these challenges, Ti implants could release Ti ions and particles into the surrounding tissues [7]. In a canine model, numerous Ti particles were detected in alveolar bones around Ti implants at 3 months, with an average diameter of 2~3 µm [8]. Martini et al. reported that at 3 months after being implanted into the ship tibiae, Ti particles could migrate to 200~500 µm from the Ti implant surface [9]. It has been reported that approximately 205~210 µg of Ti ions/particles were degraded from each gram of the implant and accumulated into the surrounding bones following 3 months of functional loadings [7]. Further, the leached Ti ions and particles negatively influenced the viability and activity of bone marrow mesenchymal stem cells (BMSCs), osteoblasts, and fibroblasts, aggravating the pro-inflammatory response with immune cells [10].

Ti cytotoxicity toward BMSCs has been demonstrated [11]. Indeed, by disturbing the cytoskeleton of BMSCs, Ti particles could alter their migration and differentiation capabilities. Furthermore, a high concentration of Ti particles (0.15–1%) affected osteoblastic viability through the excessive secretion of interleukins (Il-6 and Il-8) by neutrophils that were abnormally recruited [12,13,14,15]. For example, Mine et al. observed that the viability of murine osteoblastic cells (MC3T3-E1) was lowered by high Ti concentrations (e.g., 20 mg/L), while concentrations in the range of 1–9 mg/L were ineffective [16]. Similarly, a significant decrease (more than 50%) in osteoblast proliferation was reported when cells were exposed to Ti (48 mg/L for 3 days or 24 mg/L for 7 days) [17]. In addition, a 20% reduction of osteoblasts’ metabolic activity was observed following a 7-day exposure to 24 mg/L of Ti, and the metabolism dropped to very low levels when a higher dose (48 mg/L) was used. Interestingly, Ti shaping had an impact. Ti discs made from metal powder displayed no cytotoxicity, whereas Ti powder was cytotoxic towards human osteoblast-like cells (SAOS-2) in a time- and dose-dependent manner (at concentrations above 15.5 μg/L) [18].

Regarding bone-resorbing cells, Ti particles promote osteoclasts’ differentiation and resorption activity by enhancing the release of RANKL and pro-inflammatory cytokines (such as Il-1 and TNF-β) by macrophages and lymphocytes [19]. Osteoclasts reinforce osteoclastogenesis and are also involved in Ti surface corrosion, leading to an accumulation of metal ions in the cytoplasm and the nuclear heterochromatin of precursor cells. For example, more than 20% of 22 healthy donors displayed monocytes differentiated into mature osteoclasts after 10 days’ exposure to 48 μg/L of Ti [20]. 

To summarize, osteolysis generated by mature osteoclasts surrounding Ti implants may contribute to aseptic loosening in orthopedics [21]. This phenomenon is exacerbated by (i) an alteration of osteoblastic cell viability [22] and (ii) a pro-inflammatory response due to interactions between Ti particles with immune cells, including macrophages and T lymphocytes [23,24,25].

For implant/dental tissue interactions, Ti particles have been observed in the surrounding areas, such as the submucosal plaque, peri-implant soft tissues and bone, and in distant locations, such as lymph nodes [26,27,28]. For example, the Ti level in oral mucosa cells increased by a factor of 3 after 30 days of dental implant treatment [27]. After 3 months, a 1.4-fold increase in gingiva Ti content was found in patients with dental implants [28]. This metallic debris could migrate through the different layers of peri-implant soft tissues and generate inflammatory responses [10,29,30,31]. Histological analysis of tissues adjacent to dental implants harvested from patients revealed peri-implantitis lesions and varying grades of inflammation in the mucosa [30]. Therefore, although the clinical implication of biocorrosion and Ti particle accumulation in dental tissues is unclear, this is an area of potential concern. It is prudent to institute measures that minimize material release from a functioning implant [21].

In summary, the release of Ti ions/particles from Ti-based dental/orthodontic implants attributed to corrosion can compromise tissue integration. Studies have reported several nano-engineering techniques that modify the surface of Ti implants to significantly enhance their corrosion resistance and reduce the release of ions/particles (Figure 1). This comprehensive review introduces the degradation of ions/particles from Ti implants, the related mechanisms and contributing factors, and their influences on the surrounding cells and tissue. Further, various nano-engineering strategies for enhancing the corrosion resistance of Ti implants have been detailed, including the deposition of nanostructures, grain refinement, and EA. A particular focus of the review is anodized Ti dental implants that enable local elution of potent therapeutics to achieve augmented implant integration (both osseo and soft-tissue integration).

## 2. Factors Influencing Ti Implant Corrosion

### 2.1. Key Factors

#### 2.1.1. Mechanical Corrosion

This is observed during all the steps of dental intervention. For example, the bone-cutting instruments required for implant bed preparation cause the release of Ti particles due to metal attrition, wear, and corrosion. Ti particles and ions have been found in irrigation liquid collected during the implant bed preparation through drill or piezosurgery procedures [6]. Note that sterilization of cutting tools can initiate a corrosion process leading to particle generation [32]. Insertion of implants in the mandible and maxilla also causes friction with bone tissue, leading to the release and accumulation of Ti particles in periprosthetic tissues [33].

Similarly, implant fixing on the abutments facilitates particle release from the most fragile material [34]. However, these particles can remain in place, causing frictional wear or migrating to adjacent tissues [35]. These phenomena can be amplified by micromotions resulting from functional loading, and this can cause a mismatch between the abutment and the implant. During chewing, a more significant gap favors micromovements and fretting at the interface, which amplifies implant destabilization [36]. This space can then be colonized by microorganisms and glycoproteins, forming a stable biofilm responsible for the microbiological corrosion of the implants [37].

#### 2.1.2. Chemical and Electrochemical Corrosion

As previously described, a stable Ti oxide layer forms spontaneously on the surface of the implants. This film is exposed to functional stimuli (i.e., micromotions, micromovement of the implant–abutment interface) and various environmental conditions (i.e., acidic pH, electrolytes) that progressively alter this protective layer and thereby expose the bulk material to the oral environment [35]. Wet corrosion of dental implants is mainly seen because the oral cavity is a moist environment. Depending on its composition, pH, buffering power, and surface tension, saliva can play the role of an electrolyte which contributes to the dissolution of this oxide layer. In addition, microorganisms present in biofilms (such as *Streptococcus mutans* and *Candida albicans*) produce acidic metabolites (i.e., citric acid, lactic acid) that accelerate corrosion processes and implant failure [38,39].

#### 2.1.3. Tribocorrosion

Tribocorrosion has been defined as a combination of tribological (i.e., wear and fretting) and corrosive (i.e., chemical or electrochemical reactions) phenomena [37,40]. This term encompasses the concepts of oral environment and variations in mechanical stresses (loading, speed) impacting the integrity of dental implants. For example, implant maintenance (cleaning, disinfection) constitutes a risk of tribocorrosion. Chemical decontamination methods can damage the Ti layer and induce corrosion because of the pH. Moreover, since mechanical methods are often required to improve decontamination efficiency, Ti release can be amplified by friction, leaving the implant surface exposed. Finally, cleaning protocols via laser treatment can also be detrimental to the implant surfaces [41].

### 2.2. Specific Factors Linked to the Oral Cavity

#### 2.2.1. Mechanical Factors

As mentioned, fretting corrosion occurs when two surfaces interact by generating small oscillating movements. These micro/macro movements permanently or discontinuously aggravate the Ti implant’s degradation, as observed during mastication.

#### 2.2.2. pH Variations

The oral cavity environment fluctuates in terms of salt concentration, temperature, oral flora, oxygen content, food, drink, and tobacco consumption. Any conditions that can lower the pH below 6 favor the corrosion process. These constant aggressions lead to the degradation of the thin layer of Ti oxide and the leaching of metallic debris and ions from the surface [42,43].

#### 2.2.3. Oral Contamination

Dental implants are exposed to an environment that contains various microorganisms, and the roughness and the geometry of the implant surface favor bacterial colonization. Lipopolysaccharides (LPS) from bacteria facilitate ion exchange between saliva and Ti, affecting the thin film’s corrosion resistance. Moreover, the presence of LPS of bacterial origin is known to stimulate the activity of monocytes and macrophages responsible for the inflammation that develops in the peri-implant tissues [44]. Furthermore, inflammatory episodes occurring in the surrounding tissues can also impair the resistance of implants to corrosion processes.

#### 2.2.4. Fluoride Treatments

Most toothpastes or gels contain fluoride at concentrations between 0.1 and 2% (*w*/*v*), which can induce thin layer dissolution [45,46]. Addressing this problem, Kaneko et al. characterized the compounds formed on the implant’s surface and documented the presence of Ti fluoride, Ti oxide fluoride, and Ti sodium fluoride [47]. By replacing the original Ti oxide coat, these Ti/fluoride compounds form a more soluble layer that undergoes chemical changes leading to accelerated dissolution.

## 3. Enhancing Corrosion Resistance

### 3.1. Alloying and Chemical Modification

#### 3.1.1. Alloying of Titanium

Titanium alloys, including Ti-Zr and Ti-Nb, are routinely used to fabricate dental and orthopedic implants. Compared with the thin, amorphous TiO_2_ layer formed on pure Ti, the passive oxide film on these Ti alloys is composed of the electrochemically inert ZrO_2_ and Nb_2_O_5_, which augments their chemical stability [48,49,50,51]. It has been reported that the pitting corrosions on Ti-Zr alloys (Zr = 30~50 wt%) from acidic artificial saliva were significantly reduced compared to those on pure Ti [48]. However, the chemical stability of Ti-Zr alloys was not continuously enhanced with the increasing Zr contents. Han et al. reported that Ti-Zr with 15 wt% Zr was more electrochemically stable than Ti-Zr with 20 wt% Zr [49]. Further, incorporating Nb to obtain Ti-Nb alloys also showed enhanced chemical stability compared to pure Ti, which increased with Nb composition (5 to 10 wt%) [50]. However, the chemical stability of Ti-Nb was compromised with the presence of excessive Nb incorporation (>20 wt%), attributed to the formation of β and ω phase Nb with suboptimal stability [50]. Similar results were obtained by Çaha et al., who found Ti-15Nb (Ti-Nb with 15 wt% Nb) exhibited significantly decreased corrosion current (i*_pass_*), and a lower coefficient of friction (COF) value than Ti-40Nb, confirming its enhanced resistance against chemical corrosion and tribocorrosion [51]. In summary, alloying the Ti implants with Nb and Zr could modify its oxide layer and reduce degradation. However, ion leaching could still be detected from the Ti alloy surfaces that pose cytotoxicity concerns [49,51].

#### 3.1.2. Chemical Coatings and Modifications

To effectively inhibit the degradation from Ti implant surfaces, various chemical modifications have been employed. Hydroxyapatite (HA) is a chemical coating option that is morphologically and chemically similar to human bones. Besides promoting osseointegration, HA-coated implants were also reported to enhance the resistance against chemical corrosion by showing significantly increased polarization potential in phosphate buffer solution (PBS) [52]. Further, the corrosion current of HA-coated Ti implants was reduced considerably within Hank’s solution, indicating their reduced degradation speed [53]. Additionally, micro-arc oxidization (MAO) enables the incorporation of various ions and functional groups onto the TiO_2_ surface. For instance, MAO was utilized to incorporate phosphorus ions on Ti implants, resulting in the formation of TiP-TiO_2_ crystalline surfaces that thickened the TiO_2_ oxide layer and filled the surface cracks [54]. Compared with the non-modified Ti, the degradation speed of TiP-TiO_2_ coated Ti implants in simulated body fluid (SBF) was significantly reduced by presenting significantly fewer pits and defects after SBF immersion [54].

Further, electrochemical impedance spectroscopy (EIS) supported the enhanced corrosion resistance of the TiP-TiO_2_ crystalline surface on MAO-treated Ti implants, showing significantly increased polarization values [55]. Additionally, plasma immersion and ion implantation (PIII) could induce non-metallic ions into Ti implant surfaces to enhance their resistance against chemical corrosion [56,57,58]. Nitrogen, oxygen, and carbon have been introduced onto Ti implants via PIII to obtain a chemically modified surface that effectively shields the underlying Ti implants against chemical and electrochemical corrosion [56,57,58].

In summary, the reasons for Ti implant degradation could be categorized into mechanical corrosion, chemical/electrochemical degradation, and tribocorrosion. These corrosions are generated by functional loadings and chemical agents from the surrounding human bone and oral cavities and are aggravated by under-fluctuating pH, fluorine infiltration, bacterial infection, and inflammation [48,51,54]. Strategies for alloying or surface chemical modifications (chemical coatings, ion implantation) have been employed on Ti implants, showing enhanced chemical stability with reduced corrosion and degradation.

### 3.2. Implant Nano-Engineering

Fabrication or deposition of nanostructures on Ti implants has been explored to enhance their surface bioactivity via tailored nanotopography [59]. Strategies such as nitriding, plasma spraying of nanoparticles, and depositing nanowires have enabled the fabrication of controlled nanotopographies on Ti implants that shield underlying Ti against corrosion.

#### 3.2.1. Nitriding for a Nanoparticular TiN Layer

Nitriding diffuses nitrogen on metal surfaces via a hydrothermal treatment that facilitates a nanocrystal TiN layer to protect the underlying Ti implants [60]. Studies have reported that a nitride-treated Ti surface has improved hardness and chemical resistance, attributed to the covalent bonding of Ti-N that establishes a ceramic TiN layer. The traditional TiN coating layer was obtained by the physical vapor deposition (PVD). However, the resultant TiN coating was inconsistent and had numerous defects that can cause corrosion-based degradation. To address this shortcoming, Kazemi et al. utilized plasma-assisted chemical vapor deposition (PACVD) to obtain a consistent TiN layer on the Ti implant surface that significantly reduced the degradation of Ti within SBF, as confirmed by reduced corrosion current value [61].

Selective laser melting (SLM) can fabricate a uniform TiN layer by depositing TiN particles onto Ti implants. The SLM process also enables the refinement of TiN particles into nanoscale, densely and evenly distributing them on the Ti implants in a combination of ε-Ti_2_N and δ-TiN phase [62]. The resultant nanoparticular TiN layer significantly reduced the corrosion current density (6.92 × 10^−5^ A/cm^2^ compared with 4.64 × 10^−4^ A/cm^2^ of pure Ti) while increasing the corrosion potential (−0.126 vs. −0.238 V of pure Ti) in the acidic hydrochloric acid (HCl) solution (Figure 2) [62]. An alternative option for nitriding Ti implants is DC magnetron sputtering, which fabricates a nanocrystalline TiN protective layer with approximately 2 μm thickness [63]. The sputtered TiN nanocrystalline layer significantly reduced the degradation of Ti implants by increasing the corrosion potential (−0.837 to −0.546 V after coating) within the sodium chloride (NaCl) 3.5% solution [63].

Additionally, glow-discharge ion nitriding has also been utilized to synthesize a nanoscale roughened TiN protective layer (1.5~5 μm thickness) consisting of nitrides δ (TiN) and ε (Ti_2_N) that significantly augmented the corrosion potential till 300 mV (−150 mV for non-coated Ti) [64]. In summary, nitriding via PVD or SLM increases the corrosion resistance of Ti implants by yielding a nanocrystalline TiN layer onto implants. However, the application of Ti-nitriding is mainly restricted to aircraft/marine equipment. Hence, further investigations on its bioactivity performance are needed to enable Ti-nitriding to enhance implants’ corrosion-resistance.

#### 3.2.2. Plasma Spraying of Nanoparticles (NPs)

Plasma spraying involves melting the material powder into the plasma jet under high temperature and spraying the melted plasma to produce a coating layer with NPs on the metal surface. Utilizing plasma spraying could facilitate a hydroxyapatite (HA) NPs layer with an average thickness of 10~50 μm, which could significantly increase the corrosion potential and reduce the corrosion current in Hank’s solution (Figure 3) [65]. However, numerous gaps were observed among the sprayed NPs that compromised the consistency of the HA coating layer and limited their corrosion resistance. To address this challenge, Singh et al. mixed graphene nanoplatelets (GNPs) with the HA powder to fabricate a hybrid HA-GNP coating, within which the embedded GNPs could fill the defects among HA NPs for enhanced corrosion resistance [66].

Another option to enhance chemical stability and wear resistance of Ti is spraying Al_2_O_3_-TiO_2_ (AT) NPs. Richard et al. reported that plasma spraying could establish a refined AT NP layer with an average diameter of 30~50 nm. The AT-sprayed surface showed significantly higher coefficient friction (COF) values in the tribocorrosion tests than its ZrO_2_-sprayed counterparts, indicating enhanced resistance against tribocorrosion [67]. Further, the open circuit potential (OCP) of AT-sprayed Ti within Hank’s solution was higher than that of the ZrO_2_-sprayed and non-modified Ti implants during electrochemical tests, indicating their enhanced chemical stability that reduced degradation [67]. Similar results were reported by Palanivelu et al. that NP coating obtained by plasma spraying of AT and AT-HA particles reduced the degradation of the underlying Ti by exhibiting reduced corrosion current [68]. Further, enhanced wear resistance of AT-HA-sprayed Ti implants was obtained, along with significantly reduced weight loss and crack formation after the wear test and scratch study, respectively [69].

In summary, plasma spraying could spray NPs on Ti implants to establish a protective coating layer, shielding the underlying Ti against chemical and tribological corrosion. Further, NPs such as TiO_2_ and HA could modify their surface topography and chemistry, favorably enhancing osteoblast functions. However, the delamination of such deposited surfaces might release coated NPs to the surrounding tissues and cause local toxicity, which should be further studied and optimized for clinical translation [70].

#### 3.2.3. Depositing Nanowires (NWs)

Alkali-heat treatment and electrospinning have been reported to deposit nanowires on Ti implants, which could be interconnected into nanoscale mesh structures that shield the underlying Ti against chemical corrosion [71,72]. For instance, Zhu et al. reported the deposition of nanowires on Ti implants after a sequential acid etching (0.4% HF for 30 min) and alkali-heat treatment (soaking in 10% NaOH for 15 min) [71]. NW-modified Ti implants (Ti-NW) showed a significantly higher corrosion resistance value (R_p_ = 0.35 MΩ*cm^2^) than the pure Ti counterparts (R_p_ = 0.13 MΩ*cm^2^) within 30 mM H_2_O_2_ solution (Figure 4) [71]. An alternative NW/nanomesh deposition option was electrospinning with titanium butoxide/polyvinylpyrrolidone (PVP) composite. Manole et al. reported using NWs interconnected into a dense PVP-TiO_2_ nanomesh on Ti-Zr alloys after spinning for 30 min to significantly reduce the corrosion current and increase corrosion potential within the artificial saliva [72]. In addition, the PVP-TiO_2_ nanomesh surface exhibited similar fibroblast proliferation and spreading compared to non-modified Ti-Zr alloys [72]. However, such nanowires/nanomesh might be delaminated under mechanical loading and release NPs into the surrounding tissue. Hence, the mechanical stability of such nano-engineered Ti implants should be further evaluated and optimized.

### 3.3. Refining Grain Size into Nanoscale

Typically, a pure Ti surface has been reported with microscale grains in the range of 10~40 µm, regarded as coarse-grained Ti, with suboptimal mechanical strength. It has been widely accepted that refining the coarse-grained Ti into ultrafine grain Ti (UFG-Ti, grain size = 100~1000 nm) could significantly enhance its mechanical strength by reducing the residual stresses among the grains. Besides improving the mechanical strength, studies have reported that refining grain size enhances the resistance against chemical and tribological corrosion. Grain size refinement to the nanoscale can be achieved using various strategies described below.

#### 3.3.1. Cryogenic Treatment

Cryogenic treatment via freezing the Ti implants in liquid nitrogen (185 °C) could refine the grain size of Ti implants from several µm to 10~100 nm. Bhaskar et al. reported that a UFG-Ti with an average grain diameter of 17 nm had significantly fewer ductile dimples and micro voids after fractography studies, indicating their enhanced mechanical strength [73]. Further, cryogenic-treated UFG-Ti also showed favorable chemical resistance by forming a consistent protective TiO_2_ layer to shield the underlying Ti implants against nitric acid corrosion [73]. This could be attributed to the decreased roughness caused by reduced grain size after cryogenic treatment. Reducing the Gibbs free energy within each grain (grain center = cathodic area, boundary = anodic area) augments the formation of passive oxide film [74]. Herein, UFG-Ti with nanocrystalline grains were readily covered by a consistent and stable oxide film that significantly enhanced its resistance against electrochemical corrosions [74]. Similar results were reported by Zhu et al. that the cryogenic-treated Ti implants displayed significantly lower current density (536 nA/cm^2^) than non-modified Ti (917 nA/cm^2^) during the corrosion tests in the artificial saliva [75]. Another mechanism for enhancing the corrosion resistance is the thickened oxide film created by cryogenic treatment, which could be further improved by increasing the cryogenic treatment time [76]. As reported by Gu et al., the corrosion current of 48 h-cryo-treated Ti alloys (43 nA/cm^2^) was significantly lower than that of 24 h-treated (86 nA/cm^2^) and non-modified (153 nA/cm^2^) counterparts [76]. This was attributed to the porous oxide layer (resistance value R_p_ = 4834 Ω*cm^2^ for non-modified surface) that thickened throughout the cryogenic treatment, resulting in continuously increased resistance values with the cryogenic time (24 h = 5303 Ω*cm^2^, 48 h = 7669 Ω*cm^2^) [76]. In summary, cryogenic treatment could remodel the grain size into UFG-Ti, which reduces the internal stress and yields a consistent TiO_2_ protective layer that augments mechanical and chemical stabilities.

#### 3.3.2. Surface Mechanical Attrition Treatment (SMAT)

SMAT is a severe plastic deformation (SPD) technique that significantly refines the grain structure of metals without interfering with their chemical composition. It has been reported that the SMAT could decrease the grain size to approximately 50 nm, during which the distortion and strains could split the grains into nanoscale [77,78]. Besides enhancing the mechanical strength, SMAT also improves the chemical stability of Ti implants (Figure 5) [77]. This increased corrosion potential could be attributed to the charge transfer resistance (R_tc_) value, which was significantly increased after SMAT (763~1700 × 10^3^ Ω compared with 10.56 × 10^3^ Ω for non-modified Ti) [77]. Further, the wear corrosion of the SMAT-Ti surface was also considerably reduced, attributed to the thickened passive layer that significantly reduced the cracks and delamination after the wear test [78]. Huang et al. reported that the passive film resistance (R_p_) of SMAT-Ti was considerably higher than that of non-treated Ti counterparts in both physiological saline (PS) and simulated body fluid (SBF) solutions [79]. Moreover, compared with smooth Ti, the grain sizes and boundaries were changed more significantly on roughened Ti after SMAT; hence, SMAT was more effective in enhancing the corrosion resistance of roughened Ti implants [80]. It is also indicated that remodeling the grain size could thicken the superficial TiO_2_ layer on Ti implants by SMAT, establishing a more stable protective layer against chemical corrosion [80].

In summary, SMAT is an effective strategy to augment Ti implants’ mechanical and chemical stability by releasing internal stress and fabricating a thickened oxide layer with enhanced corrosion resistance. It is also noteworthy that refining the grain size of Ti implants via cryogenic or SMAT treatments could preserve their surface chemistry with minimum influence on their biocompatibility.

## 4. Electrochemically Anodized Ti Implants

### 4.1. Anodization of Ti Implants

Electrochemical anodization (EA) is a cost-effective nano-engineering strategy capable of fabricating various nanostructures, such as titania nanotubes (TNTs) and titania nanopores (TNPs) on Ti implants [81]. Besides the bioactivity enhancements, anodized nanotopographies also influence the corrosion resistances of modified Ti surfaces. This could be attributed to the thickened protective TiO_2_ barrier with superimposed nanostructures that shields the underlying Ti [81].

Briefly, EA involves the oxidization of target Ti within a DC-operated electrochemical cell by driving anode Ti to react with the oxygen from the electrolyte and form the TiO_2_ barrier layer (BL) [81,82]. In the F-containing electrolytes, the oxidized TiO_2_ BL then reacts with F^-^ to form water-soluble [TiF_6_]^2−^, and finally, dissolve into the electrolyte to self-order hollow nanostructures, upon attainment of an anodization equilibrium [83]. Tailoring the electrolyte conditions and adjusting EA parameters (voltage, current, and time) could yield various nanostructures on Ti implants, including titania nanotubes (TNTs), titania nanopores (TNPs), and nanotemplates (NTs) [84,85]. Additionally, it has been reported that the distribution of TNTs/TNPs was influenced by the microscale topography of the underlying substrate, and dual micro-nanostructures can be fabricated via conserved underlying substrate micro-topography [83,86]. The use of electrolyte aging (repeated use of the same electrolyte to anodize non-target Ti before anodizing target-Ti) conditions the electrolyte to enable the fabrication of dual micro-rough and nanoporous structures on micro-machined Ti implants [82,83,87].

Studies have reported the advantages of anodized Ti implants, including their bioactivity enhancements, local drug loading/delivery capability, and improved corrosion resistance [88,89]. In addition, it has been observed that osteoblasts’ and fibroblasts’ proliferation and functions were improved on anodized Ti implants with distinctive nanotopographies, promoting wound healing and tissue integration [88,89]. Further, the immune responses were also attenuated on nano-engineered Ti implant [90]. Additionally, the anodized hollow nanostructures could be utilized as nanoscale reservoirs for drug loading and local release for enhanced tissue integration and antibacterial functions [91,92]. Besides the abovementioned advantages, studies also reported that anodized Ti implants exhibit improved chemical corrosion resistance, enabling long-term stability within human bones and more corrosive oral cavities, which could be attributed to the thickened TiO_2_ BL and superimposed nanotopographies.

### 4.2. Thickened TiO_2_ Barrier Layer

Anodization treatment could enhance the chemical stability of the oxide barrier layer of Ti implants attributed to the thickened TiO_2_ barrier layer (BL). Karambakhsh et al. reported increasing the anodization voltage from 10~100 V for 30 s thickened the TiO_2_ BL without forming nanostructures and significantly reduced the passive corrosion current values in varied solutions (artificial saliva, simulated acid rain) [93]. Similarly, a study from Saraswati et al. reported that anodizing in KOH-based electrolytes thickened the TiO_2_ BL from 5 to 198.6 nm/1199 nm (anodizing at 10 V/30 V) and exhibited significantly reduced current density than non-anodized Ti in the Ringer Lactate solution [94]. In contrast, Saji et al. reported that H_3_PO_4_-based electrolyte anodized Ti implants exhibited accelerated degradation speed (higher I*_corr_*) than non-modified counterparts within Ringer’s solution [95]. This was because the acidic H_3_PO_4_-based electrolytes dissolved the TiO_2_ BL during anodization, significantly reducing its thickness under anodized TNTs and corrosion resistance [95].

### 4.3. Influence of Titania Nanostructure Layer

Dimensions of anodized porous/tubular nanostructures could also influence the corrosion resistance of Ti implants. Studies have shown that anodized Ti with TNTs exhibits improved electrochemical stability compared to bare Ti. Briefly, the fabrication of 120 nm-diameter TNTs with 20 nm thickened walls provided an additional resistance value of 1.86 × 10^6^ Ω, thereby significantly enhancing the corrosion resistance of modified Ti implants [96]. Demetrescu et al. reported that anodized Ti with 120 nm-diameter TNTs showed significantly reduced corrosion current (I*_corr_* = 0.21 µA/cm^2^) compared to unmodified Ti (I*_corr_* = 7.12 µA/cm^2^) [96]. Further, the corrosion potential (E*_corr_*) value of anodized Ti significantly increased to −0.255 from −0.380 V (non-anodized Ti) within Fusayama artificial saliva (Figure 6) [96]. The results confirmed that TNTs =significantly reduced the dissolution rate of Ti implants from 0.27 to 0.0076 mm (per year) by increasing the corrosion voltage threshold and reducing the corrosion speed [96]. Similarly, Al-Saady et al. reported that TNTs (diameter 82 nm, length 3 µm) significantly reduced the corrosion current density of Ti implants from 579 to 76 nA/cm^2^ within the artificial saliva [97]. Further, the corrosion potential of Ti implants with TNTs significantly increased from −209 to −138 mV, confirming adequate protection for the underlying Ti from TNTs [97]. Besides artificial saliva, the enhanced chemical stability of TNTs was also reported in simulated body fluids (SBF), indicating their potential application as corrosion-resistant implants [98]. Further, the TNTs were also resistant to alkaline solutions. Unlike the naturally formed TiO_2_ layer on pure Ti that immediately dissolved in 1 M KOH, anodized Ti with TNTs remained stable and exhibited significantly reduced corrosion current [99]. In summary, anodization of Ti implants to fabricated TNTs can be utilized to augment their corrosion resistance, thereby reducing corrosion rates in corrosive environments.

### 4.4. Influence of Nanotube Dimensions

Dimensions of the nanotubes (diameter and length) can easily be altered by tuning anodization voltage, current, time, and electrolyte water/fluoride content [88]. The diameter of TNTs is a critical factor in determining their corrosion resistance. For example, Liu et al. compared the polarization curves of TNTs with varied diameters (22 to 59 nm, fabricated using 5–15 V) and observed that the corrosion potential of TNTs increased with diameters, attributed to BL thickening with increased anodization voltage [100]. However, the corrosion potential abruptly decreased for 20 V-anodized TNTs (diameter 86 nm), as the large diameter enhanced the contact area of TNTs with the electrolyte that accelerated electrochemical corrosion. As a result, TNTs with an average diameter of 60 nm showed optimal corrosion resistance [100]. Similarly Huang et al., reported that TNTs (diameter 55 nm, length 400 nm) showed high corrosion resistance by presenting significantly higher corrosion potential (E*_corr_*) and resistance values (R*_p_*) than bare Ti [101]. The conclusion that large-diameter TNTs have suboptimal corrosion resistance was supported by Ossowska et al., reporting that the TNTs fabricated via a two-step anodization process with larger diameter (~100 nm) had significantly inferior corrosion resistance as compared with the TNTs fabricated via a single-step anodization (~60 nm diameter) [102]. This was attributed to the large intratubular voids that enlarged the contact area of the electrolyte–TNTs interface, thereby exhibiting suboptimal corrosion resistance [102]. However, larger pore sizes for the TNPs (nanotubes with fused tops) yielded favorable chemical corrosion resistance, as reported by Uzal et al. [99]. Briefly, TNPs fabricated using 1 h anodization showed the highest R*_p_* value (the total resistance of oxide barrier and porous layer), compared with TNPs fabricated by 15 min and 2 h anodization [99]. Notably, the fused tops of TNPs reduce the intratubular voids/gaps for contact with the electrolytes, thereby increasing the anodization time from 15 min to 1 h resulting in increased corrosion resistance [99]. However, increasing the TNPs’ diameter by extending the anodization to 2 h can reduce their corrosion resistance, as increased surface porosity augments the contact area between TNPs and electrolytes [99]. Hence, increasing the anodization time to augment the TNTs’/TNPs’ diameter could increase their corrosion resistance; however, very large TNTs/TNPs exhibits suboptimal corrosion resistance. Notably, incorporating F inside the nanotubular anodic film during anodization can also contribute to the bioactivity of osteoblasts and fibroblasts [84,90]. However, the influence of F incorporation inside TNTs on its corrosion resistance performance remains unexplored.

### 4.5. Annealing of Anodized Ti

The crystalline phase of anodized Ti is also critical in influencing the corrosion resistance of the modified implants, which could be changed by annealing (heat treatment from 260 to 760 °C alters the crystallinity of the anodized Ti). Based on the annealing temperature, the amorphous TNTs could be converted into anatase (mainly 400 °C) and rutile (around 700 °C) crystalline phases. Nogueira et al. reported that TNTs annealed at 550 °C for 2 h with a hybrid anatase–rutile phase exhibited significantly reduced passive corrosion current (J*_pass_*) within the simulated body fluids (SBF) (Figure 7) [103]. This could be attributed to the crystalline transformation of TNTs during annealing that closed the open pores of TNTs, thereby reducing the contact region of TNTs–electrolyte [103]. It is noteworthy that while annealing augments the corrosion resistance of TNTs, it can compromise the integrity of TiO_2_ BL [104].

Additionally, during electrochemical tests, more Ca ions from Hank’s solution were sedimented onto the annealed TNTs to establish a protective Ca-containing crystal layer, which enhanced corrosion resistance of annealed TNTs post-implantation [105]. Moreover, continuously increasing the annealing temperature until 650 °C yielded dual anatase–rutile TNTs that improved their corrosion resistance compared to pure anatase–TNTs [104]. Hence, post-anodization annealing is recommended for achieving augmented chemical stability of anodized Ti implants.

In summary, anodized Ti implants with TNTs exhibit enhanced corrosion resistance attributed to the thickened TiO_2_ BL and the superimposed nanostructures. However, for achieving optimized corrosion enhancements, the diameter of TNTs should be controlled within 100 nm, and additional heat treatment should be performed to enable conversion into an anatase crystalline structure.

## 5. Drug-Releasing Anodized Implants towards Implant Integration

Soft-tissue integration (STI) at the transmucosal region of dental implants and osseointegration at the implant screw-bone level is crucial to the long-term success of dental implants [1,106]. Various physical, chemical, biological, and therapeutic modifications have been performed on conventional dental implants to augment STI and osseointegration [107]. Further, therapeutic intervention may be needed in patients with ongoing conditions (diabetic, smokers and aged patients) to achieve timely establishment and long-term maintenance of implant integration [7]. For instance, suboptimal soft-tissue sealing at the transmucosal region can cause microbial ingress into the implant structures, thereby resulting in infection [108]. Electrochemically anodized Ti with nanotubes and nanopores allows for the incorporation and local elution of potent therapeutics, growth factors or proteins directly inside the implant micro-environment [109]. This ensures maximum and tailored (patient-specific) therapy, bypassing systemic administration and ensuring timely implant integration. Next, we discuss critical research advances for anodized TNTs and TNPs to achieve local drug delivery towards orchestrating STI and osseointegration.

### 5.1. Enhanced Soft-Tissue Integration

Timely establishment of soft-tissue integration (STI) is critical for the long-term success of dental implants that protects the underlying implant structures against bacterial ingress [1,110]. However, the STI on conventional dental implants may be suboptimal (especially in compromised patient conditions), requiring bioactivity enhancements on the implant surface [1,111]. Anodized Ti implants with controlled TiO_2_ nanotopography (nanotubes and nanopores) augment the activity of epithelial cells and fibroblasts towards achieving enhanced STI [88]. Local elution of specific proteins and growth factors from nanotubular Ti implants is emerging as a promising implant modification strategy to achieve accelerated STI. Next, we discuss critical studies encompassing drug-eluting anodized nano-engineered dental implants.

Wei et al. reported loading of 25~50 ng of C-terminal connective tissue growth factor fragment (CCN2) into the anodized Ti implants with nanotubes (diameter 100~120 nm, length 400~500 nm) via physical deposition [112]. An average of 76~81% of pipetted CCN2 protein solution was successfully loaded into the nanotubes and showed a continuous releasing pattern until 120 min [112]. Compared with the bare TNTs, the CCN2-loaded TNTs promoted the early-time adhesion of human skin fibroblasts and increased their proliferation rates from 1~5 day in vitro [112]. Additionally, fibroblasts’ cell–cell interaction and cell–nanotube anchorage was improved on CCN2-loaded TNTs (Figure 8) [112].

Fibroblast growth factor-2 (FGF-2) could be incorporated into nanotubes to improve fibroblast functions and enhance soft-tissue sealing. Ma et al. reported the incorporation of FGF-2 immobilized Ag nanoparticles inside TNTs to achieve upregulated proliferation and extracellular matrix (ECM) secretion from human gingival fibroblasts (hGFs) [113]. The activity of hGFs on FGF-2 loaded TNTs was influenced by the concentration of the FGF-2 solution. TNTs loaded with 500 ng/mL FGF-2 solution were reported to be optimal for the proliferation and expression of vascular endothelial growth factor (VEGF) from fibroblasts, which was higher than that of TNTs loaded with 250 ng/mL and 1000 ng/mL [113].

Besides growth factors, calcium phosphate (CaP) nanoparticles (NPs) were also deposited into TNTs to improve the activity of epithelial cells and fibroblasts [114]. Xu et al. reported that CaP NPs could be deposited into TNTs via electrochemical deposition, which significantly altered the surface chemistry of TNTs while maintaining their topography [114]. Further, the TNTs deposited with CaP-NPs were more hydrophilic than bare TNTs. Modified TNTs augmented the proliferation of human gingival epithelial cells (hGECs) and human gingival fibroblasts (hGFs) and enhanced their secretion of human epidermal growth factor (hEGF) and type I collagen, respectively [114].

Notably, oral biofilm could compromise the STI by interfering with the functioning of peri-implant epithelial cells and fibroblasts and amplifying the inflammatory responses to delay tissue healing [115,116]. Hence, to improve STI, studies evaluated the loading of antibiotics and antibacterial ions into the nanotubes to inhibit bacterial ingress. Zhao et al. reported that Ag-NPs’ incorporation into the TNTs via physical deposition exhibited an extended antibacterial capacity against *S. aureus* till 30 days [116]. However, the initial burst release of Ag-NPs can cause local toxicity to the surrounding osteoblasts and fibroblasts, which must be prevented [116]. Moreover, clinical antibiotics could also be loaded inside TNTs for their local elution. Lin et al. utilized gentamicin-loaded TNTs and found that 120~200 nm diameter TNTs could accommodate higher amounts of gentamicin to inhibit the proliferation of *S. aureus* and *S. epidermis* until 24~48 h [115]. Further, compared with the bare Ti, TNTs and gentamicin-loaded TNTs promoted the activity of mesenchymal stromal cells (MSCs) as confirmed by enhanced proliferation and ECM generation [115]. Besides Ag, gallium (Ga) is an alternative antibacterial metal ion that could be doped into the TNPs to inhibit bacteria activity [117]. Recently, Jayasree et al. reported that immersion of TNPs into 1 mg/mL Ga(NO_3_)_3_.4H_2_O solution yielded nanoscale Ga deposition onto the TNPs, showing a consistent in vitro Ga ions release for 10 days [117]. Compared with the bare TNPs, Ga-doped TNPs augmented the in vitro proliferation of hGFs at 3~7 days, and effectively reduced the in vitro viability of oral biofilm at days 1~7 [117].

### 5.2. Enhanced Osseointegration

It has been well established that the formation of living bone-implant contact (osseointegration) involves the formation and maturation of newly formed bone around Ti implants, which requires 4 weeks and 8~12 weeks on micro-roughened Ti implants, respectively. Accelerating the bone healing and regeneration ensures the long-term success of Ti implants. While anodized Ti with TNTs or TNPs augments osteoblast functions to orchestrate osteogenesis, for compromised patient conditions, further accelerated osseointegration is needed. As a result, local elution of potent orthobiologics via nanotubular Ti implants is emerging as a promising strategy.

Potent bone forming growth factors (GFs) like bone morphogenetic protein (BMP) can be loaded into TNTs for promoting bone regeneration and repair. Zhang et al. utilized BMP-2 incorporated on lentiviral vectors (Lenti-BMP-2) and localized into TNTs (50 nm diameter and 70 nm length) [118]. Therapeutic nanotubes showed a constant in vitro release of Lenti-BMP-2 until 6~8 days [118]. Compared with bare TNTs, Lenti-BMP-2 loaded TNTs significantly enhanced the expression of Runx2, osteopontin (OPN), and osteocalcin (OCN) from bone marrow stem cells (BMSCs) [118]. Further, both the Lenti-BMP-2 loaded TNTs, and bare TNTs reduced the tumor necrosis factor-α (TNF-α) and interleukin-1β (IL-1β) from RAW 264.7 macrophages, indicating favorable immunomodulation [118]. Enhanced osseointegration effect of BMP-2 loaded TNTs was also supported by in vivo placement that obtained significantly higher bone-implant contact (BIC) around BMP-2 loaded TNTs, with higher alkaline phosphate and collagen I expressions within the surrounding bone tissue [119]. 

Besides BMP, basic fibroblast growth factor (bFGF) could also be incorporated into the TNTs, via the linking of polydopamine (PDA) and heparin [120]. Compared with bare TNTs, bFGF-loaded TNTs augmented the osteogenic differentiation of dental pulp stem cells (DPSCs) by increasing the expression of alkaline phosphate (ALP), Runx2, OPN, and OCN (Figure 9) [120]. In addition, Gulati et al. reported loading of parathyroid hormone (PTH) into the TNTs fabricated on Ti wires, which significantly upregulated the expression of RANKL and suppressed the expression of SOST genes from SaOS-2 osteoblasts in a 3D cell culture model in vitro [121]. Further, to show the controlled release of therapeutics from TNTs, chitosan coating was utilized to cap the indomethacin (anti-inflammatory drug)-loaded TNTs, which reduced the initial burst release (IBR) of indomethacin from 75 to 25% within the initial 6 h and extended the drug delivery time from 3 to 18 days [121].

Attributed to favorable osteogenic capacity, calcium phosphate (CaP) nanoparticles are an alternative option to augment osseointegration of TNTs/Ti implants. Roguska et al. loaded CaP-NPs into the TNTs via immersion in Hank’s solution, yielding CaP loading inside TNTs and CaP crystals deposition on the surface of TNTs [122]. CaP deposition significantly improved the hydrophilicity of TNTs and promoted osteoblasts’ proliferation and adhesion [122]. Besides physical immersion, CaP could also be loaded into the TNTs by the alternative immersion method (AIM) [123]. Briefly, TNTs/Ti implants are immersed sequentially into the electrolytes containing Ca^2+^ and PO_4_^3−^, to yield a cluster of nanoscale needles, which significantly improved the adhesion and proliferation of MC3T3-E1 osteoprogenitor cells [123]. Chemical sputtering can also be used to dope CaP on TNTs, ensuring the successful loading of CaP NPs and a uniform CaP coating layer [124]. Chemically sputtered Ca NP-doped TNTs significantly improved the early-stage adhesion of osteoblasts and enhanced filopodia stimulation [124].

It is noteworthy that uncontrolled initial burst release (IBR) or high concentrations of locally released therapeutics or nanoparticles can cause cytotoxicity [125]. As a result, various strategies have been employed to reduce the IBR of loaded drugs, such as using a biopolymer coating on drug-loading TNTs that also offer enhanced osteogenic functions [126]. Various investigations relating to controlling the release of drugs from TNT-based Ti implants via nanotube modification, polymer coating, micellar encapsulation, or triggers are reviewed elsewhere [91]. For therapeutic NP-incorporating TNTs, intentional (therapeutic) or unintentional (breakdown or wear) release of NPs can cause cytotoxicity [70]. Mechanical failure or corrosion of the TNTs can also result in the release and generation of metal ions or NPs from TNTs and/or their therapeutic modifications. Note that NPs’ cytotoxicity can be controlled by altering the size or concentration of NPs, as described elsewhere [70].

## 6. Conclusions and Future Directions

Titanium-based dental implants experience corrosive environments within the surrounding human tissues and the oral cavity, which can cause the elution of cytotoxic nanoparticulates from the implant surface. An enhanced corrosion resistance could ensure the long-term safe functioning of Ti implants, and this review highlighted various nano-engineering surface treatments of Ti implants to achieve corrosion protection. However, several critical research gaps must be addressed to ensure the clinical translation of such corrosion-resistant nano-engineered implants.

Nitriding enables surface modification of a nanostructured TiN layer with exceptional chemical stability; however, the bioactivity performance of TiN surfaces needs further investigation. Further, plasma spraying and alkali-treatment deposits nanoparticles and nanowires to obtain a nanostructured layer that shields against corrosive electrolytes. However, their mechanical stability remains a research gap, especially considering the load-bearing implant setting. Additionally, alkali-heat treatment can alter Ti implants’ chemistry and crystallinity, influencing implants’ bioactivity and biomechanical performance.

Refining Ti implant surface grain size via cryogenic and surface mechanical attrition treatment (SMAT) is also reported to augment corrosion protection. Electrochemically anodized Ti implants with nanotubes and nanopores have recently emerged as modifications that impart multi-functionalities, including enhanced bioactivity, local drug release, and corrosion resistance. Noting that corrosion protection performance is influenced by the diameter of the nanotubes/nanopores, how this alters the bioactivity or drug loading/releasing abilities needs further investigation. However, like other nano-engineering strategies, the bioactivity of these modifications needs to be investigated in long-term implant settings in vivo under mechanical loading.

Corrosion resistance is a critical feature of titanium-based dental implants that reduces surface delamination/degradation post-implantation in the corrosive oral cavity. Various nano-engineering treatments have been employed to augment Ti implants’ corrosion resistance, ranging from nanostructure deposition (nanowires, nanocrystals), and grain size refinement to electrochemical anodization (fabrication of TiO_2_ nanotubes/pores). Such nanoscale modifications alter the Ti implant surface topographically and chemically and enable the formation of a protective TiO_2_ layer that shields the underlying Ti from corrosive electrolytes. However, whether such corrosion-resistant modifications influence the implants’ bioactivity performance and mechanical stability remains unexplored. Especially for anodized nanostructures, the implant modifications’ mechanical and corrosion resistance performance needs to be investigated in long-term *in vivo* settings. The next generation of Ti-based implants should possess high corrosion resistance without negatively impacting implant bioactivity and biomechanics.

Titanium nitriding, plasma treatment, and nanowire deposition can enhance the corrosion resistance performance of Ti-based implants. It would be interesting to couple these techniques with anodization (for instance, anodization of nitride/plasma-treated Ti) to determine if the dual modification influences corrosion protection.

Clinical translation challenges exist for anodized drug-releasing Ti implants, as summarized elsewhere [108]. These include investigating local therapy in vivo in compromised patients, the mechanical stability of nanostructures under loading, potential cytotoxicity from initial burst release of therapeutics and sterilization/packaging/shelf-life considerations. Anodized Ti with controlled nanotopography holds significant promise as the next generation of corrosion-resistant and therapeutic dental implant modification.

The future of titanium dental implants will involve nano-engineered surfaces that enable enhanced bioactivity, local therapy, and high corrosion protection without eliciting any local cytotoxicity. This can only be achieved if the identified research gaps are addressed in the coming years to ensure the clinical translation of such advanced dental implants.

## Figures and Tables

**Figure 1 pharmaceutics-15-00315-f001:**
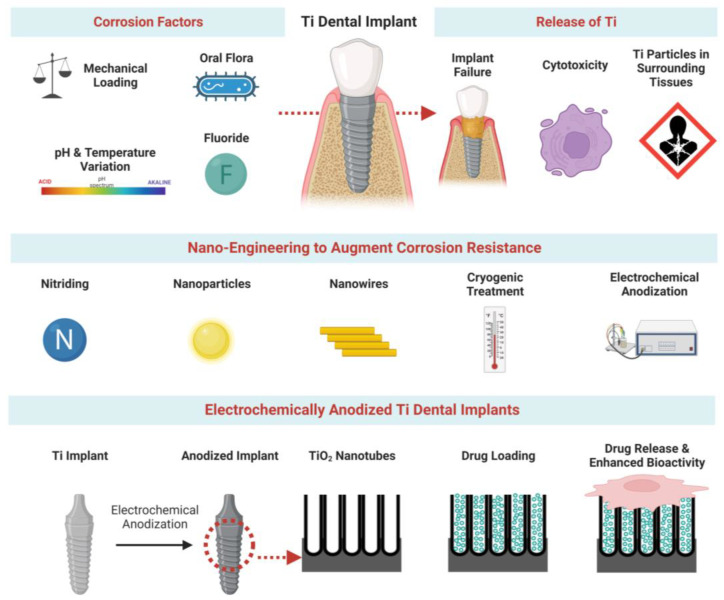
**Corrosion and local drug delivery from Ti dental implants.** Schematic representation showing corrosion factors, nano-engineering to augment corrosion resistance, and anodized Ti dental implants towards drug release and enhanced bioactivity.

**Figure 2 pharmaceutics-15-00315-f002:**
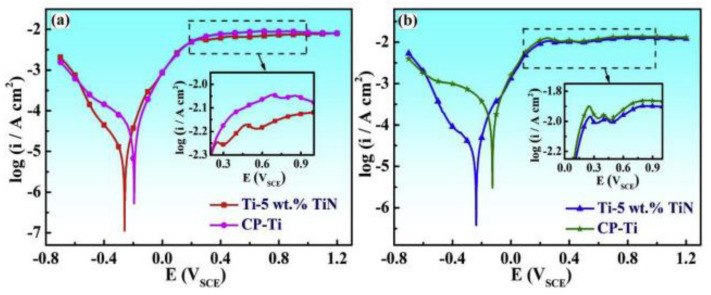
**Nitriding treatment enhances the corrosion resistance of titanium.** The polarization curves of non-modified pure Ti (CP-Ti) and nitride-treated Ti implants with 5 wt% of TiN composite (Ti-5 wt% TiN) showed that the corrosion potential of nitrided Ti implants was significantly higher than that of CP-Ti post-immersion in 0.5 M HCl solution for (**a**) 0 days and (**b**) 15 days. Reproduced with permissions from [62].

**Figure 3 pharmaceutics-15-00315-f003:**
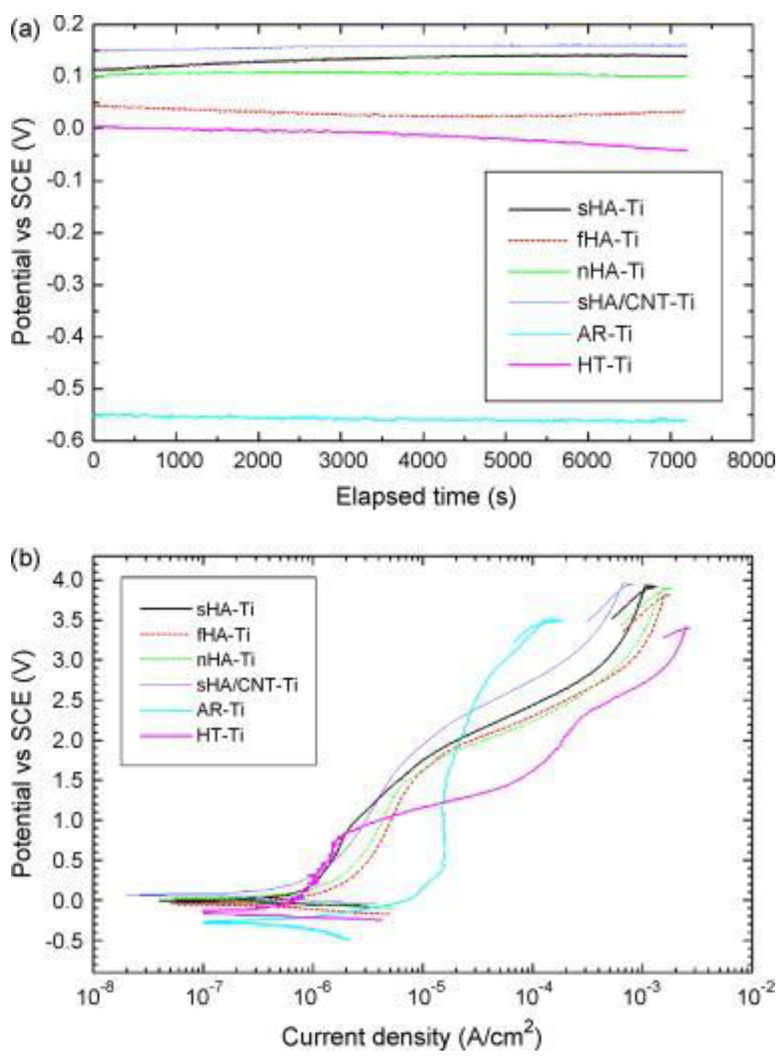
**The enhanced corrosion resistance of Ti implants modified with hydroxyapatite (HA) nanoparticles (NPs).** (**a**) The open circuit potential (OCP) and (**b**) potentiodynamic polarization curves of non-modified Ti implants and various hydroxyapatite (HA)-coated Ti implants in Hank’s solution. All the HA-coated Ti implants showed significantly higher corrosion potential than the non-coated counterparts (as-received Ti (AR-Ti) and hydrothermal-treated Ti (HT-Ti)). Reproduced with permission from [65].

**Figure 4 pharmaceutics-15-00315-f004:**
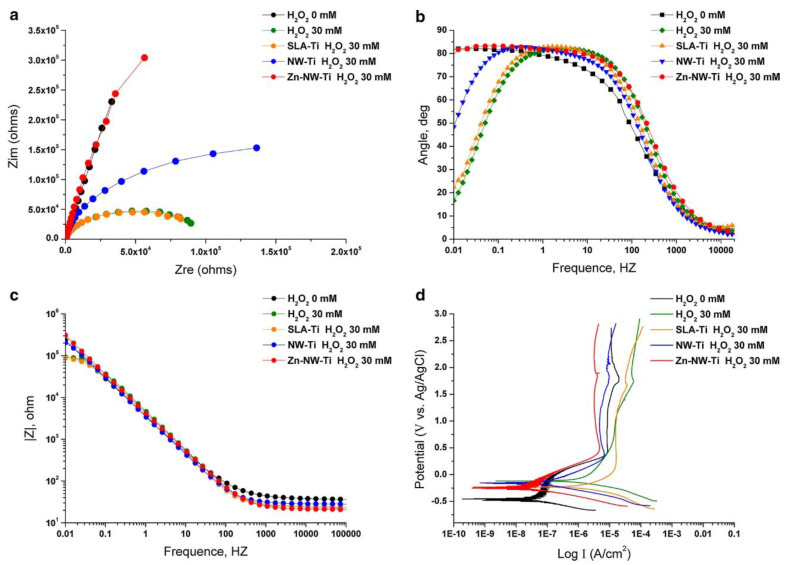
**Coating nanowires significantly enhances the corrosion resistance of Ti implants.** The electrochemical impedance spectroscopy (EIS) of pure Ti, SLA-modified Ti (SLA-Ti), TiO_2_ nanowire-coated Ti (NW-Ti), and Zn-TiO_2_ nanowire-coated Ti (Zn-NW-Ti) within 30 mM H_2_O_2_ solutions. (**a**–**c**) The Nyquist plot, Bode Phase, and Bode diagrams show enhanced corrosion resistance of NW-Ti and Zn-NW-Ti. (**d**) The potentiodynamic polarization for varied materials indicates the increased corrosion potential of NW-Ti and Zn-NW-Ti in H_2_O_2_ solutions. Reproduced with permission from [71].

**Figure 5 pharmaceutics-15-00315-f005:**
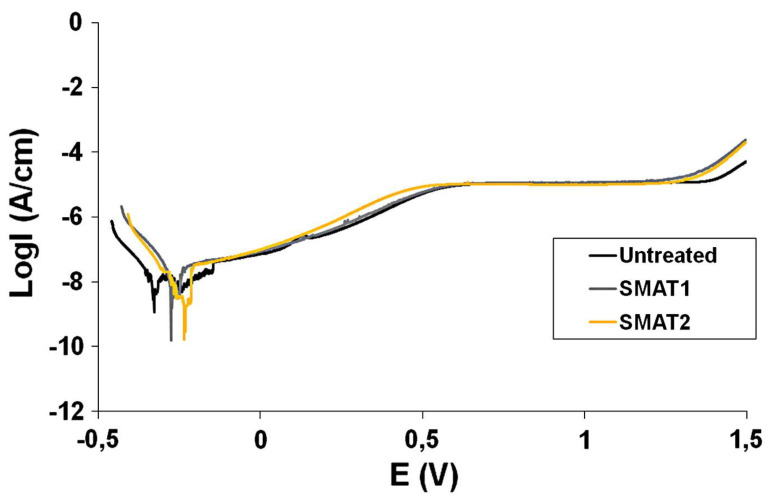
**The enhanced corrosion resistance of Ti implants modified with surface mechanical attrition treatment (SMAT).** The potentiodynamic polarization curves reveal that SMAT significantly increased the corrosion potential (E*_corr_*) while decreasing Ti implants’ corrosion current (I*_corr_*), enhancing their corrosion resistance. Reproduced with permission from [77].

**Figure 6 pharmaceutics-15-00315-f006:**
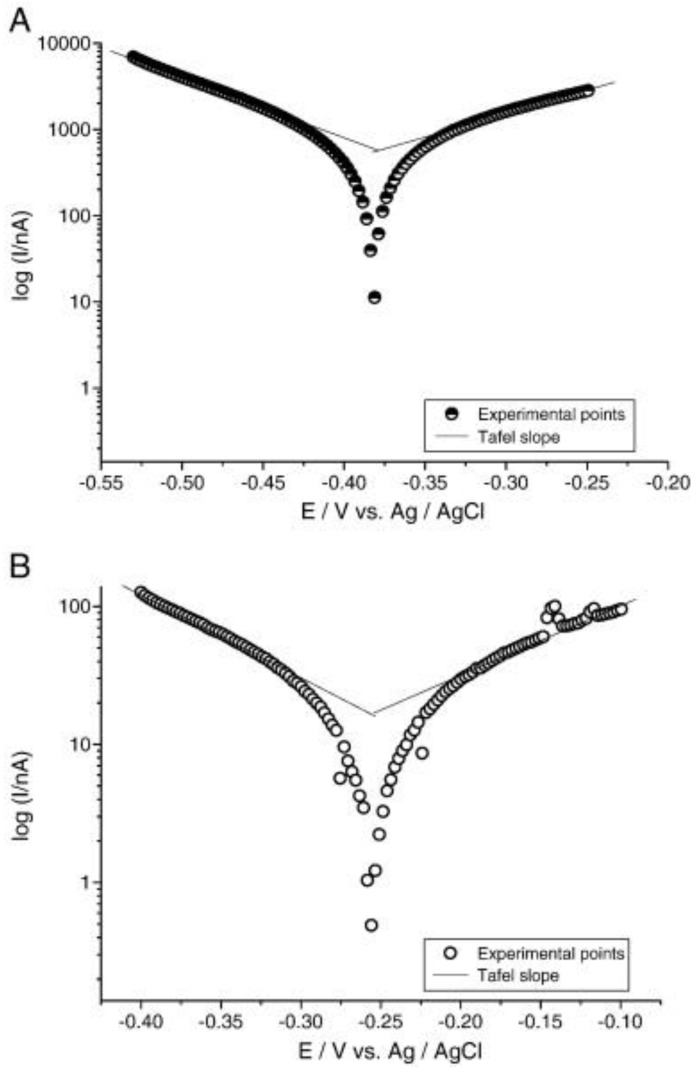
**The corrosion resistance of anodized Ti with titania nanotubes (TNTs).** Tafel plots of (**A**) TNTs modified Ti implants; and (**B**) bare Ti implants within artificial saliva. Results reveal that corrosion potential (E*_corr_*) of anodized Ti implants was significantly higher than that of the bare Ti. Reproduced with permission from [96].

**Figure 7 pharmaceutics-15-00315-f007:**
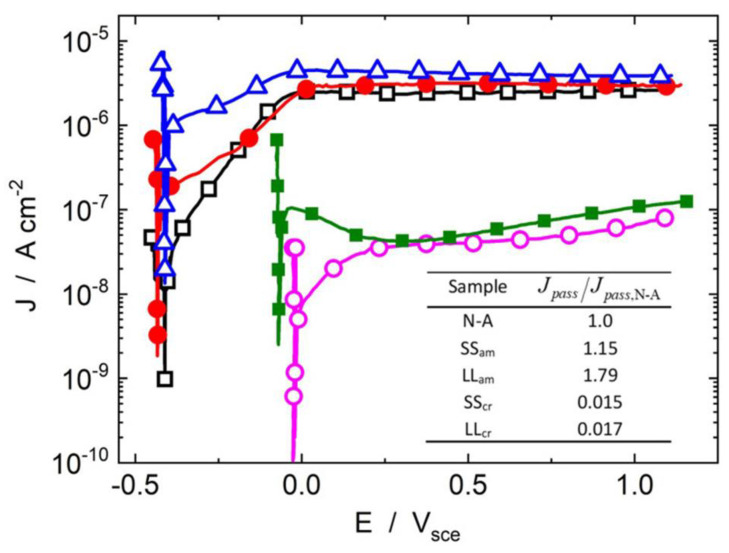
**Influence of annealing of anodized Ti on corrosion resistance.** Potentiodynamic polarization curves of non-anodized Ti implants (N-A: black), anodized Ti implants with amorphous TNTs (SS_am_: red, LL_am_: blue), and annealed TNTs with anatase-rutile crystalline phase (SS_cr_: pink, LL_cr_: green). Results show that the annealing treatment significantly reduces the passive corrosion current (J*_pass_*) of TNTs, while increasing their corrosion potentials. Inset: passivation current densities [J*_pass_*, normalized against the non-anodized sample, J*_pass,N-A_*]. Reproduced with permissions from [103].

**Figure 8 pharmaceutics-15-00315-f008:**
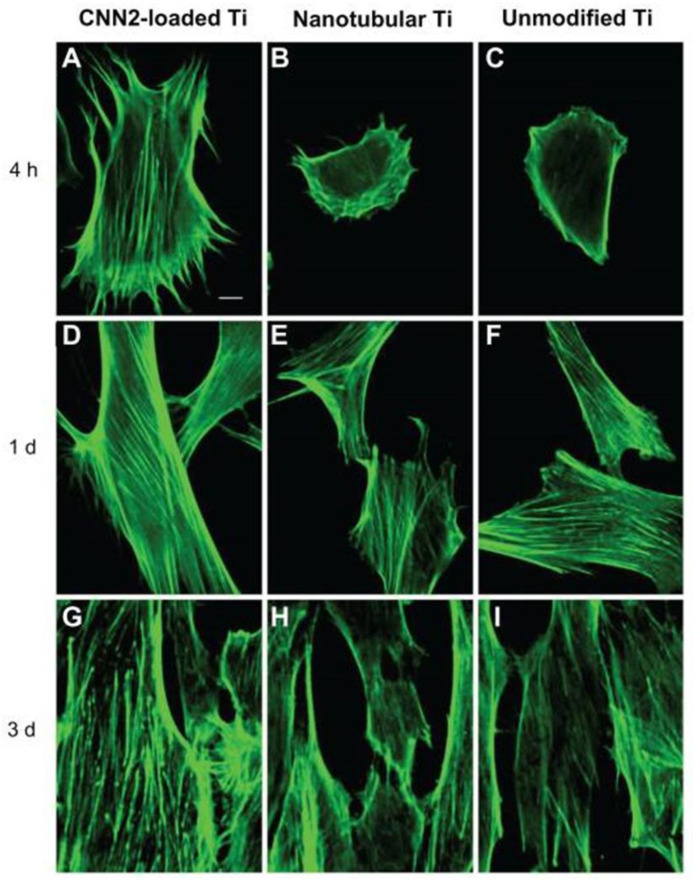
**Protein-releasing nanotubular implants towards soft-tissue integration.** Confocal imaging of interaction of human skin fibroblasts on CNN2-loaded, nanotubular and bare Ti implants. (**A**–**C**) Cell morphology was more stretched with additional stress fibers on CCN2-loaded nanotubes at 4 h. (**D**–**I**) Higher amount of intracellular contacts with confluent stress fibers were observed on CCN2-loaded nanotubes. Reproduced with permission from [112].

**Figure 9 pharmaceutics-15-00315-f009:**
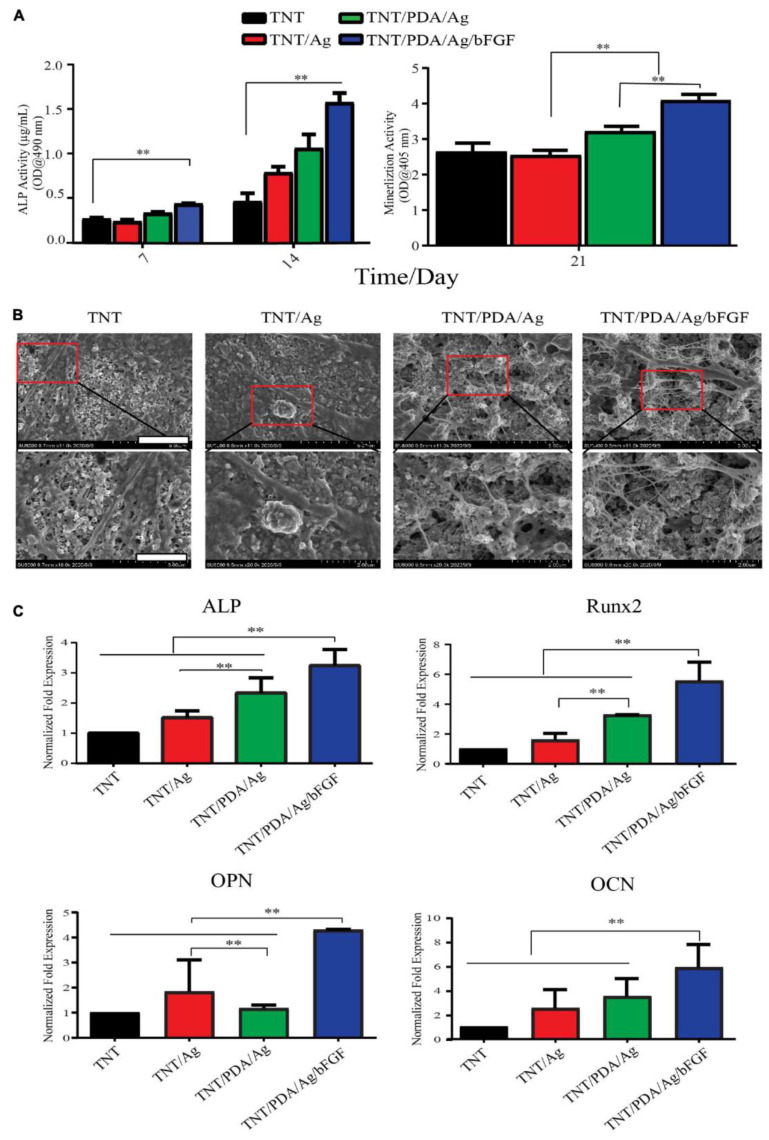
**Growth factor-releasing nanotubular implants towards osseointegration.** Enhanced osteogenic differentiation of dental pulp stem cells (DPSCs) on bFGF-loaded TNTs. (**A**) DPSCs showed significantly higher alkaline phosphate (ALP) activity; (**A**,**B**) more calcium deposition on bFGF-loaded TNTs, detected by Alizarin red staining (**A**) and SEM observation (**B**). (**C**) Significantly higher expression of ALP, Runx2, osteopontin (OPN), and osteocalcin (OCN) from DPSCs. ** *p* < 0.01. Reproduced with permission from [120].

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
