# Peer review of "Enhanced Corrosion Resistance and Local Therapy from Nano-Engineered Titanium Dental Implants"

_pharmaceutics, 2023, doi:10.3390/pharmaceutics15020315_

Round 1
Reviewer 1 Report
An excellent and thorough review of dental and orthopedic uses of titanium and clinical issues regarding cytotoxicity. Some minor changes in English grammar, word use, and sentence structure are needed. The reviewed copy of the manuscript provides suggestions.

Author Response
Reviewer #1
An excellent and thorough review of dental and orthopedic uses of titanium and clinical issues regarding cytotoxicity. Some minor changes in English grammar, word use, and sentence structure are needed. The reviewed copy of the manuscript provides suggestions.
Authors’ Response:
We would like to thank the Reviewer for their feedback and for recognizing the study's potential. We have amended the manuscript based on edits and suggestions performed by the Reviewer. Further, a thorough check of grammar, word use and sentence structure has been performed throughout the manuscript.
Reviewer 2 Report
The presented article is a well-written review, which clearly demonstrates the potential solution for enhancing the corrosion resistance of Ti-based implants, preventing Ti ions from releasing.
I recommend the paper for publication in its present form.
However, I have a recommendation to the Authors about the title of the review. I would suggest modifying it because the topic presented is wider than in the title.
Author Response
Reviewer #2
The presented article is a well-written review, which clearly demonstrates the potential solution for enhancing the corrosion resistance of Ti-based implants, preventing Ti ions from releasing.
I recommend the paper for publication in its present form.
However, I have a recommendation to the Authors about the title of the review. I would suggest modifying it because the topic presented is wider than in the title.
Authors’ Response:
We would like to thank the Reviewer for their recommendation and feedback.
We have changed the manuscript title and removed the word ‘Anodized’ from it. The new title is ‘Enhanced Corrosion Resistance and Local Therapy from Nano-Engineered Titanium Dental Implants’.
Reviewer 3 Report
Manuscript titled 'Enhanced Corrosion Resistance and Local Therapy from Anodized Nano-Engineered Titanium Dental Implants' by Guo et al. details the current trends and research gaps relating to nanoscale modified Ti dental implants and their corrosion resistance and local drug delivery performance. The review is up-to-date and addresses a crucial gap in the current Ti-based dental implants. Please address the following comments:
- Anodization of Ti also enables Fluoride incorporation inside the anodic film, how does that influence cell functions and corrosion?
- Authors discuss cytotoxicity of Ti ions or nanoparticles from Ti dental implants, what about cytotoxicity concerns relating to very high drug release amounts or the release of metal-based therapeutic nanoparticles? How can it correspond to corrosion?
- Line no. 458-460 and reference [45] are not matching. Please recheck reference citations. A recent review on the topic is available (10.1002/tcr.202200053)
- Are there any studies that combine anodization with approaches described in Section 3.1; 3.2.1; 3.2.2; etc. Like anodization of nitride modified or plasma treated Ti, that improves corrosion performance? Will that be considered a research gap to direct future studies?
- As per the author's opinion, what is the future in this field? What is the recommendation?
- Are there any currently clinically used anodized implants (in dentistry and other fields)?
Author Response
Reviewer #3
Manuscript titled 'Enhanced Corrosion Resistance and Local Therapy from Anodized Nano-Engineered Titanium Dental Implants' by Guo et al. details the current trends and research gaps relating to nanoscale modified Ti dental implants and their corrosion resistance and local drug delivery performance. The review is up-to-date and addresses a crucial gap in the current Ti-based dental implants. Please address the following comments:
Authors’ Response:
We would like to thank the Reviewer for recognizing the potential of our manuscript. We have addressed all comments from the Reviewer as detailed below.
- Anodization of Ti also enables Fluoride incorporation inside the anodic film, how does that influence cell functions and corrosion?
Authors’ Response:
Besides the morphology of nanostructures, the F incorporation during anodization can positively influence the proliferation of fibroblasts [Guo et al. Appl Surf Sci, 2021] and osteoblasts [Gulati et al. Mater Sci Eng C, 2018]. However, most corrosion studies on anodized Ti have been restricted to the thickened anodic oxide film and dimensions of nanotubes.
The related discussions have been added to Page 14, Line 524-527.
- Authors discuss cytotoxicity of Ti ions or nanoparticles from Ti dental implants, what about cytotoxicity concerns relating to very high drug release amounts or the release of metal-based therapeutic nanoparticles? How can it correspond to corrosion?
Authors’ Response:
Cytotoxicity due to high release of drugs or therapeutic NPs from Ti dental implants is a major concern and we have included new discussion in this regard [Page 20, Line 694-705]. The audience is directed to relevant reviews with additional details on controlling drug release (which is out of scope of the current study). Corrosion can cause unintended release of Ti ions or NPs, however, influence of intentional release of therapeutic NPs on corrosion is not yet explored.
- Line no. 458-460 and reference [45] are not matching. Please recheck reference citations. A recent review on the topic is available (10.1002/tcr.202200053)
Authors’ Response:
Thanks for pointing out this oversight. We have now cited the correct reference in that sentence [new reference number 95]. Further, the paper that the reviewer suggested is also cited [reference number 125].
- Are there any studies that combine anodization with approaches described in Section 3.1; 3.2.1; 3.2.2; etc. Like anodization of nitride modified or plasma treated Ti, that improves corrosion performance? Will that be considered a research gap to direct future studies?
Authors’ Response:
We thank the Reviewer for pointing this out. Please note that there are studies where micropatterned Ti (for example via laser treatment) followed by anodization to fabricate TNTs have promoted the mechanical stability of TNTs onto the Ti substrate [https://doi.org/10.1002/adfm.200801703]. However, combination of nitriding, plasma or nanowire deposition and anodization to augment corrosion performance remains unexplored. We have now included additional details on this in Section 6 (Page 21, Line 747-750).
- As per the author's opinion, what is the future in this field? What is the recommendation?
Authors’ Response:
Section 6 (Conclusions and Future Directions) describes the current challenges, research gaps and future recommendations relating to nano-engineered Ti dental implants to achieve superior corrosion protection. Briefly, the following passage summarizes our recommendation:
The future of titanium dental implants will involve nano-engineered surfaces that enable enhanced bioactivity, local therapy and high corrosion protection without eliciting any local cytotoxicity. This can only be achieved if the identified research gaps are addressed in the coming years to ensure the clinical translation of such advanced dental implants.
This recommendation is now included at the end of Section 6 (Page 21, Line 757-761)
- Are there any currently clinically used anodized implants (in dentistry and other fields)?
Authors’ Response:
We thank the Reviewer for this query. Please note that electrochemically anodized Ti dental implants with controlled TiO2nanotopography (such as nanotubes or nanopores) have not been clinically utilized to-date.